# Diagnosis of Migrainous Infarction: A Case Report and Analysis of Previously Published Cases

**DOI:** 10.3390/diagnostics13152502

**Published:** 2023-07-27

**Authors:** Elena R. Lebedeva, Natalia M. Gurary, Jes Olesen

**Affiliations:** 1Department of Neurology, The Ural State Medical University, 620028 Yekaterinburg, Russia; 2International Headache Centre “Europe-Asia”, 620144 Yekaterinburg, Russia; 3Medical Union “New Hospital”, 620028 Yekaterinburg, Russia; 4Danish Headache Centre, Department of Neurology, Rigshospitalet-Glostrup, University of Copenhagen, 1172 Copenhagen, Denmark

**Keywords:** migrainous infarction, migraine with aura, diagnostic criteria, International Classification of Headache Disorders

## Abstract

Migrainous infarction (MI) is a rare disorder. The precise diagnostic criteria for this diagnosis have been available in the International Classification of Headache Disorders (ICHD) since 1988, but many cases do not fulfil these criteria. This paper describes a good example of MI and analyzes previously published case reports. We report a very special case of MI in an 18-year-old woman who had a recurrent episode of migraine with a similar aura with numbness of the right arm and speech disturbances which had an unusually long duration (>120 min). On admission to the headache centre “Europe-Asia”, she complained of slowness of speech and problems with choice of words. An MRI showed acute lacunar infarcts in the left parietal subcortical area. Ischemic infarcts were localized in a relevant area on the left side and the aura symptoms were right-sided. The patient, therefore, fulfilled the ICHD-3 diagnostic criteria for “Migrainous infarction”. An analysis of 35 previously published articles with case reports about MI showed that 22 did not meet the diagnostic criteria of the ICHD for migrainous infarction. Using all this, we developed diagnostic recommendations for migrainous infarction which can help to improve the quality of diagnosis when used together with the diagnostic criteria of the ICHD for migrainous infarction.

## 1. Introduction

Migrainous infarction (MI) is a rare disorder that reportedly accounts for 0.5–1.5% of all ischemic strokes [1]. It is more frequent in females than in males, especially before the age of 50. MI mostly occurs in the posterior circulation. It was first defined in the International Classification of Headache Disorders (ICHD) in 1988 [2]. The latest version, ICHD-3, gave the following description of migrainous infarction: one or more migraine aura symptoms occurring in association with an ischaemic brain lesion in the appropriate territory demonstrated by neuroimaging, with onset during a typical migraine with aura attack [3]. The diagnostic criteria for MI of ICHD-3 are presented in Figure 1.

Precise diagnostic criteria for this diagnosis are thus available, but many case reports did not fulfil these criteria and MI has therefore been used with varying meanings in the literature. The pathophysiology of MI is unclear. The proposed possible mechanisms include cerebral vasospasm, cerebral embolism, right to left cardiac shunts, prolongation of the migraine aura itself and cortical spreading depression, hemodynamic changes, enhanced platelet aggregation, endothelial dysfunction, genetic syndromes, and predisposition [1,4,5]. Here, we report on a young woman fulfilling the ICHD-3 criteria for migrainous infarction and analyse previously published cases.

## 2. Case History

### 2.1. Patient Information

An 18-year-old woman (a student) was admitted to the International Headache Centre “Europe-Asia” in Yekaterinburg on 4 April 2018 for evaluation and treatment of a headache with numbness of the right arm and speech disturbances. The previous day, she had experienced an episode of right-sided visual disturbances (photopsias followed by scotoma) followed by paraesthesia and numbness of fingers of the right hand spreading to all of the right hand, right half of the face and tongue. This was followed by speech disturbances (“slurred” speech). All symptoms spread gradually and one followed the other. They continued for 60 min except for speech disturbances which persisted for more than 120 min. These symptoms were accompanied by a headache that occurred 30 min after the onset of focal neurological symptoms. The headache was bilateral, pressing, of moderate intensity, without aggravation by physical activity, photo- or phonophobia, vomiting or nausea and was unresponsive to combined analgesics and triptans. Its duration was around 12 h. No triggers for the neurological event were identified. The patient gave informed concern and written permission to use her MRI scans for this article.

### 2.2. Clinical Findings

#### The First Visit

On admission to the headache centre, she complained of decreased vision, problems with concentration and choice of words. She had experienced, six months earlier, two similar episodes of headache accompanied by visual, sensitive and speech disturbances which continued for around 60 min.

In addition, she had a history of unilateral or bilateral headaches in the frontotemporal regions since 17 years old. They had moderate intensity, were pressing or pulsating, lasted 4–6 h, and were sometimes accompanied by photo- and phonophobia, but no nausea or vomiting and no aggravation by routine physical activity. The frequency was one per 2 months.

Upon admission, slowness of speech was detected in her neurological status. A physical examination showed no abnormalities. Her vital signs included a blood pressure of 100/70, a heart rate of 70 beats per minute, respirations of 16 per minute, and she was afebrile. She did not have a history of hypertension and cigarette/alcohol use, no family history of migraine or other headache, and no family history of stroke. Her cholesterol and glucose levels were normal. No vascular risk factors were identified. The diagnosis was “Migraine with aura and migraine without aura”.

### 2.3. Diagnostic Assessment and Evaluation

MRI was performed two days later on 6 April 2018. It showed acute lacunar infarcts in the left parietal subcortical area (Figure 2).

MR-angiography of cerebral vessels and cervical MRI with fat suppression were unremarkable. A duplex scanning of the cervical arteries detected hypoplasia of the right vertebral artery but no abnormalities of the carotid arteries. CT-angiography of the cerebral and neck vessels was normal. Transcranial Doppler cerebral embolus detection found 24 microembolus signals 7 dB in the left middle cerebral artery. For the detection of a potential source of embolism, we made the following examinations: (1) contrast-enhanced echocardiography with a transesophageal approach; (2) peripheral arterial and venous Doppler ultrasonography; (3) 24 h monitoring of electrocardiogram (EKG); and (4) computed tomography (CT) of lungs. All these examinations were normal. Lipids, glucose, and coagulation were normal. No mutations in the NOTCH 3 gene and no antiphospholipid antibodies were detected. Vasculitis workup included the following tests: protein C, protein S, antithrombin III, fibrinogen, d-dimer, clotting factors II, VII, and not organ-specific antibodies (anticardiolipin, antithyroglobulin, Lupus anticoagulant). An ophthalmologist performed visual field testing and found no abnormalities.

In summary, MRI detected acute lacunar infarcts in the left subcortical area and parietal lobe. The patient had had two previous attacks of migraine with aura and the present attack had a longer duration. An ischemic infarct was localized in a relevant area on the left side and the aura symptoms were right-sided. The patient, therefore, fulfilled the ICHD-3 diagnostic criteria for “Migrainous infarction”.

### 2.4. Treatment

Aspirin (100 mg) was prescribed to this patient for long-lasting therapy and candesartan (8 mg) for the prophylactic treatment of migraines for 12 months.

### 2.5. Follow-Up of the Patient

#### 2.5.1. The Second Visit: 27 October 2018

The patient did not have any headaches and no aura symptoms after the first visit. She complained only about slowness in writing. The neurological examination was normal. Her blood pressure was 100/70 and she had a heart rate of 72 beats per minute. She continued to use aspirin (100 mg) and candesartan (8 mg). A transcranial Doppler cerebral embolus detection did not find any embolic signals.

#### 2.5.2. The Third Visit: 23 August 2019

The patient had a third attack of migraine with aura accompanied by visual disturbances (photopsias followed by scotoma) in December 2018. She had no complaints upon admission to the headache centre. The neurological and physical examinations were normal. She continued to use aspirin (100 mg) and candesartan (8 mg). A second MRI was performed on 21 August 2019. It revealed cysts, glial changes, and leukoaraiosis in the left parietal subcortical area (Figure 3).

Monitoring for 24 h of the EKG and transcranial Doppler cerebral embolus detection did not find any abnormalities. Lipids, glucose, and coagulation were normal.

#### 2.5.3. The Fourth Visit: 21 December 2020

The patient had no complaints. A neurological and physical examination showed no abnormalities. She had stopped candesartan but continued to use aspirin (100 mg). MRI (19 December 2020) revealed no new findings compared to the previous imaging. The results of MR-angiography of the cerebral vessels, 24 h monitoring of EKG, lipids, glucose, and coagulation were normal.

#### 2.5.4. The Fifth Visit: 27 January 2022

The patient complained about attacks of migraine with visual aura (photopsias followed by scotoma) once every other month. The duration of the visual aura was 30–60 min, and the headaches started 5 min after the aura onset. She used eletriptan for the acute treatment of migraine and continued aspirin (100 mg) prophylaxis. Candesartan (8 mg) was prescribed again for 12 months. Examinations including 24 h monitoring of EKG, transcranial Doppler cerebral embolus detection, lipids, glucose, and coagulation were normal.

#### 2.5.5. The Sixth Visit: 22 November 2022

The patient continued candesartan (8 mg) and aspirin (100 mg) daily. She had no attacks of migraine with aura from January 2022 and no other complaints. Her blood pressure was normal. The neurological and physical examinations were unremarkable.

#### 2.5.6. Telephone Interview: 21 May 2023

The patient had no complaints and used both candesartan (8 mg) and aspirin (100 mg).

## 3. Discussion

The detailed clinical characteristics of headaches are key in the differential diagnosis between primary and secondary headache disorders. It is crucially important to detect their changes because they can serve as a red flag warning about a causative disorder. The main changed characteristics in the presented case were a prolonged duration of migrainous aura and aphasia. Together these symptoms were indications for MRI and MR-angiography and led to the diagnosis of migrainous infarction.

Is it migrainous infarction (MI) or was it just an ischemic stroke in a young woman? Migraine aura-like symptoms can arise at the onset of ischemic stroke [6]. However, the main distinction between these disorders is the sudden onset of focal neurological symptoms in ischemic stroke and their gradual development in MI. All symptoms had a gradual spread in the described above case and one followed the other (they occurred in succession). The symptoms included irritative symptoms and were accompanied by headaches. The patient already had a migraine with an aura typical of previous attacks except that one aura symptom persisted for >60 min. Neuroimaging demonstrated ischaemic infarction in a relevant area. Therefore, it fulfilled all the diagnostic criteria for MI.

It seems from the literature that a cerebral infarct can provoke aura but a typical aura can also lead to MI. This young woman had 24 microembolus signals in the middle cerebral artery registered upon transcranial Doppler cerebral embolus detection. We found no peripheral or central source of embolism and cardiac and extracardiac shunts and arterial disease were absent. Later, several additional examinations revealed no emboli.

Cerebral embolism can induce a migrainous aura [4,5,7,8]. Furthermore, cerebral misery perfusion can also produce a migrainous aura [9]. This is a condition with reduced regional cerebral blood flow relative to the regional metabolic demand for oxygen due to occlusive carotid disease. However, this young woman did not have atherosclerosis. So, the probable cause of prolonged aura in this case report was embolism.

One may wonder why she had an embolism and also why the embolism caused the same symptoms as her previous and subsequent auras. Perhaps she had small asymptomatic emboli in several brain regions but one region responded with a cortical spreading depression, the region that before and after the infarct also spontaneously produced cortical spreading depression and aura symptoms. This is similar to focal epilepsy. We analyzed the results of our prospective study where a neurologist interviewed 550 stroke patients using predesigned forms allowing a precise diagnosis of previous as well as new auras/headaches around the time of stroke. A total of 73 patients from 550 stroke patients (13%) had a history of migraine; among them 8 (1.5%) had a history of migraine with aura. However, we applied the criteria for migrainous infarction to all patients at the admission to the hospital before neuroimaging. We asked patients about the presence of aura and headache at the moment of the development of stroke, their localization, site, signs, development sequence, duration of each symptom and other accompanying symptoms, presence of auras and migraine with aura and without aura in the past. In the case of the coexistence of aura at stroke onset, we carried out additional analysis of comparison of the aura site/localization and site/localization of the infarct. If ischaemic infarction was detected in a relevant area (for example, the aura was on the right side and the infarct was on the left side), we analyzed the previous aura symptoms and the number of these episodes in the past. Migraine auras typically spread in 5–20 min and each symptom does not last more than 60 min. However, in the case of two or more aura symptoms, they can even last two or more hours. If a patient had, in the past, at least two similar episodes of migraine with aura, we diagnosed migraine with aura. If a patient had typical previous attacks, except that one or more aura symptoms persisted for >60 min, we diagnosed MI. Among 550 patients with first-ever ischemic stroke, we found three cases of migraine with aura which developed at stroke onset and two of them were migrainous infarctions. These cases fully met the diagnostic criteria of the ICHD and were previously published in the Ural medical journal in Russian [10]. Here, we provide a shorter description of these cases. We observed a woman (53 years old) and a man (54 years old) with MI. Both patients had long histories of migraines with aura and without aura (over more than 30 years). In both cases, infarcts developed during the attack of migraine with aura, focal symptoms were similar to the previous aura symptoms but persisted for >60 min, and MRI revealed acute infarct in the frontal area on the left side in the first case and acute infarct in the territory of right middle cerebral artery in the second case. The migrainous infarct manifested as motor and amnestic aphasia in the first patient and the second patient had homonymous hemianopia on the right side, left-side hemiparesis and hemihypoesthesia, and dysarthria. These symptoms were permanent for 3 days in the first patient and 4 days in the second patient and disappeared after that. Both patients had the following risk factors: hypertonic diseases and minor signs of atherosclerosis of cerebral arteries. The first patient used oral contraceptives for 4 days, had an increased level of cholesterol, and body mass index > 25. The second patient smoked for more than 20 years. The follow-up period was 3 years. Neither of these patients had a recurrent stroke or transient ischemic attacks.

We critically reviewed and analyzed data from other published case reports of migrainous infarction [1,10,11,12,13,14,15,16,17,18,19,20,21,22,23,24,25,26,27,28,29,30,31,32,33,34,35,36,37,38,39,40,41,42,43]. In case they did not meet the ICHD criteria for MI, we wrote comments and made an expert diagnosis. The data are summarized in Table 1.

Thus, out of 35 published articles about migrainous infarction case reports, 22 did not meet the diagnostic criteria of the ICHD for MI. The most frequent causes were the following: (1) not enough information about characteristics of past auras, including the number of migrainous auras; (2) focal symptoms at stroke onset were different from previous attacks of migraine with aura; (3) the patient had migraines without aura before ischemic stroke but not migraines with aura; or (4) the absence of full description of cerebral infarct including their localization. The main problem was the absence of a detailed comparison of previous aura symptoms and focal symptoms at the onset of cerebral infarction. We summarized all this in the following recommendations for improvements in diagnosing migrainous infarction.

## 4. Diagnostic Recommendations for Migrainous Infarction

(1)It is important to ask a patient who has a focal neurological deficit about present and previous headaches and their characteristics, and to classify headaches according to the International Classification of Headache Disorders.(2)It is necessary to know the duration of focal neurological symptoms and how they developed: sudden onset, all symptoms at the same time or gradually, one after the other. Were they positive or negative?(3)The sudden onset of negative focal neurological symptoms warns of TIA or stroke.(4)The gradual spread of focal neurological symptoms with positive/irritative signs and headache are diagnostic for migraine with aura except for cases with long duration of focal symptoms (>60 min) which require urgent neurological examinations, imaging including MRI with DWI, and other investigations for the early diagnosis of cerebral infarcts.(5)A neurologist must compare current and previous clinical characteristics of headache and aura symptoms at stroke onset, as well as their site/localization in relation to the site/localization of acute cerebral infarct, in order to differentiate migrainous infarction and ischemic stroke from aura-like symptoms.(6)It is necessary to perform neuroimaging in patients with migraine with aura when they have abnormal neurological status, atypical features, or “red flags”, and this should be used for the early detection of migrainous infarction.(7)The diagnostic criteria of migrainous infarction of the ICHD should be used by neurologists in daily practice for improvement in diagnoses.

## 5. Conclusions

The described case reports of migrainous infarction and the analysis of previous cases of migrainous infarction in the literature allowed us to develop diagnostic recommendations for migrainous infarction which can help to improve the quality of diagnosis when used together with the diagnostic criteria of the ICHD for migrainous infarction.

## Figures and Tables

**Figure 1 diagnostics-13-02502-f001:**
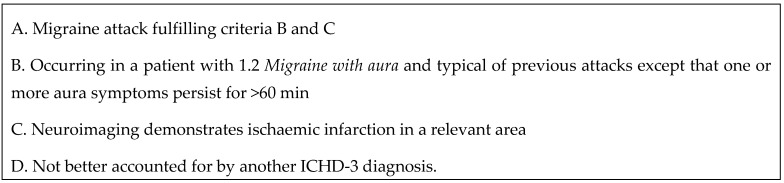
Diagnostic criteria of migrainous infarction of the International Classification of Headache Disorders, third edition (ICHD-3).

**Figure 2 diagnostics-13-02502-f002:**
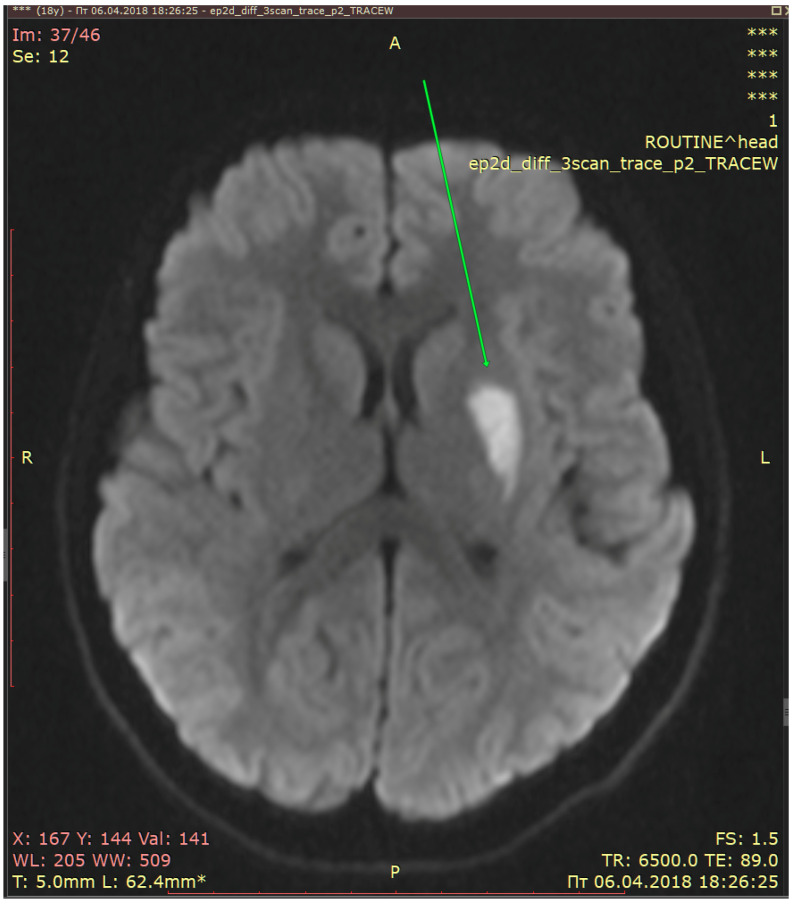
Acute lacunar infarcts (arrow) in the left parietal subcortical area (MRI, DWI).

**Figure 3 diagnostics-13-02502-f003:**
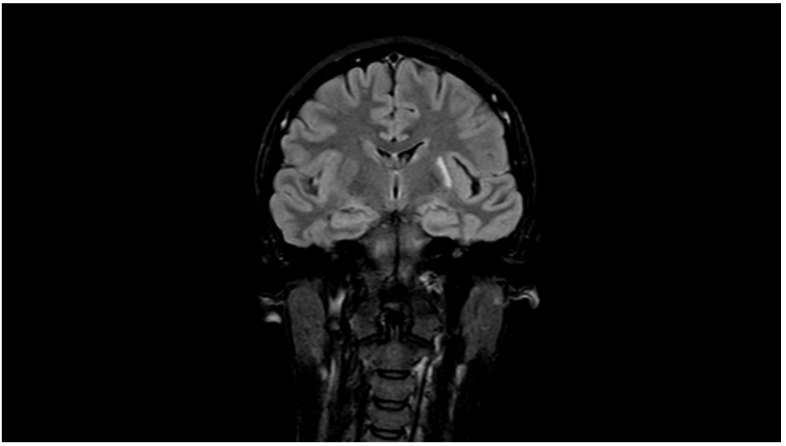
Brain changes 1 year after migrainous infarction: cysts, glial changes, and leukoaraiosis in the left parietal subcortical area (MRI, FLAIR).

**Table 1 diagnostics-13-02502-t001:** Analysis of previously published cases of migrainous infarction (MI).

*N*	Authors, Year	Number, Age, and Sex	Diagnosis Fulfils the Criteria of MI in the ICHD (Yes or No)	Comments and Expert Diagnosis
1	Lebedeva et al. (2023, present publication)	1/female/44 y.o.	Yes	Migrainous infarction
2	Vinciguerra et al. (2019) [1]	1/female/44 y.o.	Yes	Migrainous infarction
3	Mancini et al. (2019) [12]	1/male/32 y.o.	Yes	Migrainous infarction
4	Campagna et al. (2018) [13]	1/female/47 y.o.	Yes	The exact duration of the migrainous aura is not indicated in migrainous infarction
5	Khardenavis et al. (2018) [14]	1/female/27 y.o.	Yes	Migrainous infarction
6	Morais et al. (2018) [15]	1/female/37 y.o.	Yes	Migrainous infarction
7	Serrano et al. (2018) [16]	8 females and 7 males/18–55 y.o.	Uncertain	Absence of detailed characteristics and number of migrainous auras in the past and at the onset of infarction in all cases, absence of full description of all infarcts
8	Kreling et al. (2017) [17]	1/female/16 y.o.	No	Migrainous aura was not similar to the previous migraine with visual aura and had features of basilar aura
9	Renard et al. (2015) [18]	1/male/47	Yes	Migrainous infarction
10	Lebedeva at all (2015) [10]	2/1 female 53 y.o, 1 male 54 y.o.	Yes	Migrainous infarction
11	Parks et al. (2014) [19]	1/female/59 y.o.	No	Absence of infarction upon MRI with DWI
12	Thissen et al. (2014) [20]	1/female/74 y.o.	No	Persistent aura with infarction
13	Arboix et al. (2013) [21]	1/female/29 y.o.	No	Focal neurological symptoms (hemiparesis, left hemihypesthesia, dysarthria) were not similar to the previous migraine with visual aura and had features of hemiplegic aura
14	Lai e Hong (2012) [22]	1/male/60 y.o.	Yes	Migrainous infarction
15	Wolf et al. (2011) [23]	4 males and 13 females/20–71 y.o.	No	Not enough information about the characteristics of aura
16	Laurell et al. (2011) [24]	13 males and 20 females/16–76 y.o.	No	Not enough information about the characteristics of aura
17	Tsai et al. (2010) [25]	1/female/42 y.o.	No	The patient had a migraine without aura before the ischemic stroke
18	Decima et al. (2009) [26]	1/male/41 y.o.	No	Not enough information about the characteristics of aura
19	Caballero (2009) [27]	1/female/21 y.o.	No	Focal neurological symptoms (left homonymous hemianopia, metamorphopsias, dysarthria, left hemi-paraesthesias) were not similar to the previous attacks of migraine with visual aura
20	Schulz et al. (2009) [28]	3 males and 2 females/21–58 y.o.	Uncertain	Not enough information about the comparison of characteristics of aura and focal symptoms, absence of description of localization of infarcts
21	Arai et al. (2008) [29]	1/male/64 y.o.	Yes	Migrainous infarction
22	Marshall et al. (2007) [30]	1/female/57 y.o.	No	Neurological symptoms at the onset of stroke (delirium, right arm weakness, near total visual loss) were not similar to the previous attacks of migraine with visual aura
23	Liang e Scott (2007) [31]	1/female/57 y.o.	Yes	Migrainous infarction
24	Tzoulis et al. (2006) [32]	1/female/93 y.o.	Yes	Migrainous infarction
25	Matsuo et al. (2006) [33]	1/female/35 y.o.	No	The patient had a migraine without aura before the ischemic stroke
26	Frigerio et al. (2004) [34]	1 male and 5 females/23–40 y.o.	No	Several patients had focal neurological symptoms which were not similar to the previous attacks of migraine with aura
27	Tang et al. (2004) [35]	1 male, 29 y.o. and 1 female 47 y.o.	No	Not enough information about the characteristics of aura
28	Lee et al. (2003) [36]	1 male 40 y.o. and 1 female 25 y.o.	No	MRI and MR angiography did not reveal an infarct in a male. Neurological symptoms at the onset of stroke in a woman (vertigo, subjective right-sided hearing loss, diplopia, quadriparesis, right-sided hemihypesthesia) were not similar to the previous attacks of migraine with visual aura
29	Arboix et al. (2003) [37]	3 males and 6 females/24–60 y.o.	Uncertain	Absence of detailed characteristics and information about the number of migrainous auras in the past and no comparison of previous aura symptoms and symptoms at the onset of infarction
30	Linetsky et al. (2001) [38]	1 male and 5 females/15–46 y.o.	Not all cases were migrainous infarctions	Several patients had focal neurological symptoms at the onset of stroke which were not similar to the previous attacks of migraine with visual aura and their stroke manifested mainly as motor and sensory deficits and only one patient had evident hemianopsia upon bedside examination
31	Demirkaya et al. (1999) [39]	1/male/38 y.o.	No	The patient had a migraine without aura before the ischemic stroke
32	Meschia et al. (1998) [40]	1/male/43 y.o.	No	The patient had a migraine without aura before the ischemic stroke
33	Mendizabal et al. (1997) [41]	1/female/47 y.o.	No	Focal symptoms at stroke onset were not typical of previous attacks of migraine with basilar aura since neurological examination showed severe stupor without obvious focal findings
34	Sanin et al. (1993) [42]	1/female/47 y.o.	No	Focal symptoms at stroke onset (cortical blindness and left-sided hemiparesis and hemineglect) were different from previous attacks of migraine with visual aura
35	Gomez et al. (1991) [43]	1/female/33	No	Not enough information about all characteristics of aura

## Data Availability

Information can be made available under written request.

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
