# Peer review of "Diagnosis of Migrainous Infarction: A Case Report and Analysis of Previously Published Cases"

_diagnostics, 2023, doi:10.3390/diagnostics13152502_

Round 1
Reviewer 1 Report
This is a case report of migrainous infarction, followed by a literature review and diagnostic recommendations developed by the authors. This is an interesting topic and adds to the literature. I have a few minor suggestions to improve the manuscript:
1. Title: Analysis is misspelled
2. Figure 1: There is a presumably random “1.2” in the middle of the figure.
3. Figure 2: The FLAIR sequence is shown, which is nonspecific. Consider showing DWI sequence +/- ADC sequence instead, to demonstrate evidence of acute/subacute infarction.
4. Table 1: Line 19, an extra “20” in the table that needs removal
Grammatical errors do not affect overall understanding of the topics of the text. Some minor English language editing could improve the manuscript.
Author Response
Thank you very much for your detailed analysis of our manuscript. We made changed according to your comments.
1)Title: Analysis is misspelled
Thank you very much, it was typos, I corrected.
2) Figure 1: There is a presumably random “1.2” in the middle of the figure
Thank you for your note. 1.2 Migraine with aura is the code of migraine with aura in the International Classification of headache disorders. It is necessary to include this code according the criteria of this classification.
3) Figure 2: The FLAIR sequence is shown, which is nonspecific. Consider showing DWI sequence +/- ADC sequence instead, to demonstrate evidence of acute/subacute infarction.
Thank you very much for this comment. I inserted another figure with DWI sequence.
4) Table 1: Line 19, an extra “20” in the table that needs removal
Thank you, I removed it.
Reviewer 2 Report
Authors presented an interesting case report of MI. I believe that is a very interesting case that should be reported in literature. I have thought some major and minor concern about the manuscript that I hope will increase its quality.
As major concern:
- I believe that a DWI scan should have been reported to confirm the timing of the acute ischemic lesion.
As minor concerns:
- figure 2: an arrow or a circle pointing out the cerebral infarction could be useful. Even in figure 3 authors may consider different marks pointing the alterations detected
- paragraph 2.5.3: describing MRI, authors may just say that there were no new findings compared to the previous imaging.
- discussion: in the 2nd paragraph: "the symptoms were positive", actually they were both positive (at first, i.e., photopsias or paresthesias) and negative (then, i.e., scotoma, numbness and slurred speech), as per cortical spreading depression.
- discussion: in the 4th paragraph: is there a different physiopathological mechanism between embolism and atrial fibrillation? I.e., does atrial fibrillation causes migraine aura with any other mechanism than embolism?
- discussion: in the 6th paragraph: authors give too much space to their previous prospective study on 550 stroke patients; regarding it: why the stroke patients had been investigated for auras around the time of the stroke? The symptoms may have been ischemic. Besides, did all the patients have a diagnosis of migraine? In this case the investigation could have been more appropriate. Did the 2 patients described have any previous diagnosis of migraine? They boht showed mild risk factors for stroke as well.
- The title should be corrected to "analysis" and NOT "analyzis";
- some time verbs should be at the past tense. A general overview of English style is suggested.
Author Response
We are very grateful to you for your very careful and detailed analysis of our manuscript. We appreciate it very much and made changed according to your comments.
Authors presented an interesting case report of MI. I believe that it is a very interesting case that should be reported in literature. I have thought some major and minor concern about the manuscript that I hope will increase its quality.
As major concern:
- I believe that a DWI scan should have been reported to confirm the timing of the acute ischemic lesion.
Thank you very much for this important comment. We changed Figure 1 and inserted DWI scan.
As minor concerns:
- figure 2: an arrow or a circle pointing out the cerebral infarction could be useful. Even in figure 3 authors may consider different marks pointing the alterations detected
Thank you, done.
- paragraph 2.5.3: describing MRI, authors may just say that there were no new findings compared to the previous imaging.
Thank you, done.
- discussion: in the 2nd paragraph: "the symptoms were positive", actually they were both positive (at first, i.e., photopsias or paresthesias) and negative (then, i.e., scotoma, numbness and slurred speech), as per cortical spreading depression.
Thank you for your important comment, we changed this: The symptoms included irritative symptoms and accompanied by headaches.
- discussion: in the 4th paragraph: is there a different physiopathological mechanism between embolism and atrial fibrillation? I.e., does atrial fibrillation causes migraine aura with any other mechanism than embolism?
Thank you very much for this valuable comment, we deleted atrial fibrillation since it produces embolism as well.
- discussion: in the 6th paragraph: authors give too much space to their previous prospective study on 550 stroke patients; regarding it: why the stroke patients had been investigated for auras around the time of the stroke? The symptoms may have been ischemic. Besides, did all the patients have a diagnosis of migraine? In this case the investigation could have been more appropriate.
Thank you very much for these very important comments. We added additional data. We wrote that 73 patients from 550 stroke patients (13%) had a history of migraine, among them 8 (1.5%) had a history of migraine with aura. However, we applied criteria for migrainous infarction to all patients at the admission to the hospital before neuroimaging. We aimed to answer the following questions:
- Did patient have migraine with aura before stroke?
- Did a usual aura occur at the time of stroke?
- Did stroke consist of usual aura persisting?
Did the 2 patients described have any previous diagnosis of migraine? They both showed mild risk factors for stroke as well.
Many thanks for this important note. Both patients had long history of migraine with aura and without aura (during more than 30 years).
Comments on the Quality of English Language
- The title should be corrected to "analysis" and NOT "analyzis"
Thank you, done.
- some time verbs should be at the past tense. A general overview of English style is suggested.
